# Why the Anti-Meningococcal B Vaccination during Adolescence Should Be Implemented in Italy: An Overview of Available Evidence

**DOI:** 10.3390/microorganisms8111681

**Published:** 2020-10-29

**Authors:** Sara Boccalini, Beatrice Zanella, Paolo Landa, Daniela Amicizia, Angela Bechini, Maddalena Innocenti, Mariasilvia Iovine, Elvina Lecini, Francesca Marchini, Diana Paolini, Gino Sartor, Francesca Zangrillo, Piero Luigi Lai, Paolo Bonanni, Donatella Panatto

**Affiliations:** 1Department of Health Sciences, University of Florence, 50134 Florence, Italy; beatrice.zanella@unifi.it (B.Z.); angela.bechini@unifi.it (A.B.); gino.sartor@unifi.it (G.S.); paolo.bonanni@unifi.it (P.B.); 2Department of Economics, University of Genoa, 16132 Genoa, Italy; paolo.landa@fsa.ulaval.ca; 3Department of Operations and Decision Systems, Université Laval, Quebec, QC G1V 0A6, Canada; 4Department of Health Sciences, University of Genoa, 16132 Genoa, Italy; daniela.amicizia@unige.it (D.A.); mariasilvia.iovine@outlook.it (M.I.); elvinalecini@gmail.com (E.L.); f.marchini13@gmail.com (F.M.); pierolai@unige.it (P.L.L.); panatto@unige.it (D.P.); 5AUSL Toscana Centro, 50122 Florence, Italy; maddalena.innocenti@uslcentro.toscana.it; 6Health Management Unit, Careggi University Hospital, 50134 Florence, Italy; paolinid@aou-careggi.toscana.it; 7Prevention Department, Public Health Office, ASL2 Liguria, Italy; francesca.zangrillo@live.it

**Keywords:** *Neisseria meningitidis* B, adolescents, meningococcal B vaccine, anti-MenB vaccination strategy, meningococcal disease burden, meningococcal economic burden

## Abstract

Although meningococcal disease has a low incidence in Italy, it is a public health concern owing to its high lethality rate and high frequency of transitory and/or permanent sequelae among survivors. The highest incidence rates are recorded in infants, children and adolescents, and most of the cases are due to *Neisseria meningitidis* B. In Italy, anti-meningococcal B (anti-MenB) vaccination is free for infants but, despite the considerable disease burden in adolescents, no national recommendation to vaccinate in this age-group is currently available. The aim of this study was to assess the main available scientific evidence to support the Italian health authorities in implementing a program of free anti-MenB vaccination for adolescents. We conducted an overview of the scientific literature on epidemiology, disease burden, immunogenicity and safety of available vaccines, and economic evaluations of vaccination strategies. Each case of invasive meningococcal disease generates a considerable health burden (lethality rate: 9%; up to 60% of patients experience at least one sequela) in terms of impaired quality of life for survivors and high direct and indirect costs (the mean overall cost of acute phase for a single case amounts to about EUR 13,952; the costs for post-acute and the long-term phases may vary widely depending of the type of sequela, reaching an annual cost of about EUR 100,000 in cases of severe neurological damage). Furthermore, vaccination against meningococcus B in adolescence proved cost-effective. The study highlights the need to actively offer the anti-MenB vaccination during adolescence at a national level. This would make it possible to avoid premature deaths and reduce the high costs borne by the National Health Service and by society of supporting survivors who suffer temporary and/or permanent sequelae.

## 1. Introduction

*Neisseria meningitidis* (*N. meningitidis*) is an aerobic, Gram-negative bacterium that exclusively infects the human species and is the leading cause of invasive bacterial disease in the world [1]. Of the 13 serogroups currently known, only six are able to cause invasive disease (A, B, C, W-135, Y and X) [2,3], and the distribution of serogroups varies from one geographical area to another [3].

The incidence of invasive meningococcal disease varies greatly according to the geographical area [3]. Although the disease is rare in developed countries, it imposes a heavy clinical, social and economic burden, owing to its high lethality (8–15%); in the event of sepsis, the lethality rate can reach 40% [4,5,6]. With regard to Europe, the latest report by the European Centre for Disease Prevention and Control (ECDC) (2011–2015) showed that the overall rate of lethality was 9%. Furthermore, a large number of survivors (up to 60%) suffer of transient and/or permanent sequelae that impact heavily on their quality of life and that of their family members [7].

The incidence of invasive meningococcal disease varies according to age. It is highest in infants and toddlers, with a second peak often occurring among adolescents and young people [5,6]. In the adolescent population, the symptoms are generally recognized late, thus hospitalization is prolonged and physical and psychological outcomes are more frequent and more serious [8,9]. Regarding invasive meningococcal B disease, the highest frequencies of severe clinical developments are reported among children and adolescents (0–18 years): indeed, 29–41.6% of patients require admission to the intensive care unit [10,11].

In Europe, serogroup B is the most common (> 50% of cases) [6]. This pattern can also be seen in Italy; indeed, in the 2011–2017 period, about 36% of cases were due to serogroup B [12].

In Italy, the surveillance of invasive bacterial diseases is coordinated by the Italian National Institute of Health and requires that all cases of diseases caused by *Neisseria meningitidis*, *Streptococcus pneumoniae* and *Haemophilus influenzae* be reported [12,13].

The microorganism is transmitted via the airborne route and the main source of contagion is healthy carriers. The highest rates of *N. meningitidis* carrier status are reported among adolescents and young adults. This fact substantially contributes to the spread of the infection throughout the population [14,15,16].

The most effective preventive weapon available in the fight against invasive disease due to *Neisseria meningitidis* is vaccination.

In Italy, immunization against invasive meningococcal diseases has improved enormously in the last 10 years. Anti-meningococcal C or ACYW135 vaccination is now recommended for infants aged 13–15 months and for adolescents aged 12–18 years, according to the current National Vaccination Prevention Plan 2017–2019 (NVPP 2017–2019) [17]. As a further step towards the control of meningococcal disease, the NVPP 2017–2019 underlines the priority of introducing an anti-meningococcal B immunization in early childhood, with the recommendation to administer this vaccination within the first year of life. Owing to the epidemiology of *N. meningitidis B*, the Plan also reports the need to assess the active offer of this vaccine to adolescents in the near future [17]. As a confirmation of this health priority, in the recent document entitled “Life Time Immunization Schedule”, the Italian scientific associations and medical federations suggested that anti-MenB vaccination should be provided during adolescence [18]. These indications have been accepted and implemented by three Italian Regions (Puglia, Sicily and Campania) [19,20,21,22].

Today, two vaccines against meningococcal B disease are available. The vaccine with four antigenic components (MenB-4C) can be administered to children >2 months of age with the vaccination schedule varying according to the age of subject [23]. The recombinant anti-meningococcal B vaccine with two antigenic components (MenB-FHbp) is licensed for use in subjects aged ≥10 years, with the recommendation of a 2- or 3-dose schedule according to the epidemiological and clinical risk factors [24].

The aim of this study was to highlight the main available scientific evidence to support the Italian health authorities, decision makers and stakeholders in the implementation of an anti-meningococcal B vaccination program for adolescents.

## 2. Materials and Methods

An overview of the scientific literature and an evaluation of Italian surveillance data was conducted on the following relevant issues: the epidemiology and burden of invasive meningococcal B disease and its related costs, the immunogenicity and safety of the anti-MenB vaccines, and the cost-effectiveness of vaccination.

### 2.1. Epidemiology of Invasive Meningococcal Disease in Italy

As the aim of the study was to evaluate the meningococcal B disease burden in the Italian context, we analysed the Italian surveillance data (Invasive Bacterial Disease Surveillance System) concerning the period 2011–2018 [12,13]. This time period was chosen for two reasons: (1) to obtain a picture of the current epidemiological burden of the meningococcal B disease and (2) to identify cases due to *N. meningitidis* B more accurately. The notification system has become more sensitive over the years: indeed, the causative agent (*Neisseria meningitidis*, *Streptococcus pneumoniae* and *Haemophilus influenzae*) of invasive bacterial disease was not identified in about 6% in 2014, 4.4% in 2016, 3.9% in 2017, and 2.4% in 2018 [12,13]. Furthermore, the identification of the *N. meningitidis* serogroup has become more accurate; the percentage of non-typed meningococcal cases was 16%, 9% and 7% in 2016, 2017 and 2018, respectively [13].The data were analysed according to age-group and broken down by serogroup.

### 2.2. Invasive Meningococcal Disease Burden and Related Costs

To retrieve papers that would be of use in drawing up the present study, we searched PubMed, Embase and Scopus databases. Initially, we used the following research string: (Meningitis OR meningococcal disease) AND adolescent* AND (meningococcus B OR meningococcus type B OR Neisseria meningitidis B OR Neisseria meningitidis type B) AND (complication* OR sequelae) and restricted the search to the period 2000–2018. The choice of the time period was prompted by the fact that studies carried out before 2000 reported only global data on sequelae, without indicating the types (physical, neurological, psychological sequelae); they were therefore unsuitable for the present study. Both Italian and international studies were deemed eligible, though international articles were included only if published in English. Moreover, studies carried out in countries with a high incidence of meningitis (e.g., the African meningitis belt) were excluded. Subsequently, the search was broadened as the studies conducted exclusively on adolescents and that considered the sequelae divided by serogroup were few and not exhaustive. In the secondary search, we considered studies that reported data on sequelae of meningococcal disease caused by all serogroups and in all age-classes.

A manual search was then carried out by examining the bibliographies of the papers included in the present overview, in order to bring to light any sources that were not identified through the automatic search. Subsequently, each manuscript was reinserted into the search engine of Google Scholar (www.scholar.google.it), in order to identify articles that cited the studies included. Once duplicates had been eliminated, the titles and abstracts of the articles were evaluated. Subsequently, the full texts were examined.

In addition, in order to identify studies that reported the costs related to meningococcal disease, we conducted bibliographic searches by means of the search engines PubMed, Embase and Scopus. In the first phase, particular attention was devoted to studies conducted in Italy and on adolescents; subsequently, as no exhaustive studies had evaluated the costs of meningococcal disease in Italy, our research was expanded in order to consider studies conducted in high-income countries and in all age-classes.

### 2.3. Immunogenicity and Safety of Anti-MenB Vaccines

As the aim of this study was to support the Italian health authorities in the implementation of an anti-MenB vaccination program for adolescents, we evaluated the immunogenicity and safety data of available vaccines in this age-group, as reported by the authorization details [23,24].

### 2.4. Cost-Effectiveness Analyses of Anti-MenB Vaccination in Adolescents

As the aim of the study was to evaluate globally the meningococcal B disease burden in Italy, we conducted bibliographic searches by means of the search engines PubMed, Embase and Scopus in order to identify studies that reported the cost-effectiveness analyses that aimed to evaluate the economic profile of the new immunization programme. Particular attention was devoted to studies conducted in Italy and on adolescents.

## 3. Results

### 3.1. Invasive Meningococcal B Disease: A Low-Incidence Disease with a Heavy Disease Burden

During 2012–2016, Italy has recorded lower notification rates for invasive meningococcal disease (0.2–0.4 cases per 100,000 inhabitants) than the European average (0.5–0.7 cases per 100,000 inhabitants) [5,6,12,13]. The latest available report (2016–2018) reported incidence rates that were between 0.37 and 0.28 cases per 100,000 inhabitants in 2016 and 2018, respectively; these values are lower than the European average of 0.6 cases reported in 2017 (most recent data available) [6].

The most severely affected age-groups are those aged 0–4 years, but non-negligible incidence rates are also observed in the 10–14-year age-group (average of 0.39 cases per 100,000) and in the 15–24-year age-group (0.54 cases per 100,000) (2011–2017) [12]. The latest report (2016–2018) showed incidence rates in the 10–14-year age-group of: 0.32, 0.42 and 0.24 cases per 100,000 in 2016, 2017 and 2018, respectively, and of 0.86, 0.58 and 0.51 cases per 100,000 in 15–24-year age-group, for the same years, respectively [13].

In the period 2011–2017, *N. meningitidis* B was the most frequently detected serogroup, being typed in about 36% of cases of invasive meningococcal disease. In 2018, *N. meningitidis* B was detected in 44.9% of typed cases. Considering only adolescents (10–14 years) and young people (15–24 years), the average percentage of *N. meningitidis* B, out of the total cases of invasive meningococcal disease (2011–2017), was about 28% and 32%, respectively; in 2018 it was 42.9% and 51.7% of the total typed cases, respectively [12,13].

The latest report stated that, in the period 2016–2018, the most frequent clinical presentation of meningococcal disease was meningitis (35–44% of cases), followed by sepsis/bacteremia not associated with any other clinical picture (25–35% of cases) and by meningitis associated with sepsis/bacteremia (26–30% of cases). Other clinical manifestations were rare [13].

The impact of the invasive meningococcal disease is mainly due to the transitory and/or permanent sequelae (physical, neurological and/or psychological)—of variable severity—which afflict the survivors. Indeed, it is estimated that up to 60% of patients experience at least one sequela, but they may also have multiple sequelae [7]. For example, Sadarangani et al. reported that in 37% of cases the sequelae were multiple: 33% in patients aged <18 years and 42% in adults [8].

To date, no study has analyzed the impact of sequelae due to meningococcal disease in Italy. The data presented in this study have therefore been taken from studies conducted in other high-income countries. A total of 27 articles were included in the present overview.

The percentage of subjects with complications varies according to age, the severity of the acute phase and the serogroup involved; infections caused by serogroups B and C are the most severe, as demonstrated by several studies [1,25,26]. Meningococcal sequelae can be divided into three categories (physical, neurological and psychological); the most frequently observed are reported in Table 1.

Among the physical sequelae, the percentages reported in the published studies are variable and present a broad range [1,25,26]. A Canadian study (2002–2011) reported that patients aged <18 years most frequently suffered amputations (7.6%), scars (4.3%), renal dysfunction (1.4%) and joint problems (1%) [8]. Another study, conducted in Australia in the period 2002–2011, revealed that 75.6% of the subjects aged <18 years suffered sequelae due to meningococcus B and that, among these, 32.2% had skin complications (such as necrosis or scarring), 25.8% had joint and bone problems and 6.4% had suffered amputations [11].

With regard to neurological sequelae, as shown in Table 1, the published studies report very varied data [1,25,26,27,28,29,30,31]. An Australian study confirmed the high impact of the disease caused by *N. meningitidis* B, with 25.8% of patients suffering neurological sequelae [11]. These outcomes were further corroborated by a British study, in which 20.2% of survivors of invasive meningococcal B disease suffered deafness, while 9.2% had convulsions and 5.5% memory impairment [32].

Another important aspect to consider is concerns related to psychiatric and psychological problems such as anxiety or depression [33]. As they generally ensue after hospitalization, these problems are frequently underestimated in the medium and long term [11,31]. Most survivors suffer from post-traumatic stress disorders (up to 62%); however, owing to the heavy impact of physical and neurological complications, these sequelae may easily be overlooked [32,33].

Focusing on survivors from serogroup B meningococcal disease, the study carried out by Viner et al. showed that up to 26% of children suffered from psychological sequelae (mainly anxiety and behavioral disorders) that emerged late, i.e., 3–5 years after since the acute phase [10]. Gottfredsson et al. reported that 20% of survivors from serogroup B meningococcal disease had mental problems (5.7% depression, 7.1% anxiety and 2.8% anxiety/depression); the probability of having psychiatric and psychological problems was significantly higher than in the general population [31].

Finally, the psychological sequelae that may afflict the family of a survivor also constitute a substantial health concern. Studies assessing the behaviors of parents have shown that, during the acute and the post-acute phases, about 60% of mothers and 40% of fathers have psychological or psychiatric disorders that require specialist support [34,35].

### 3.2. The High Costs of Invasive Meningococcal Disease

Each invasive meningococcal disease case generates high direct and indirect costs.

To date, no exhaustive studies have evaluated the global costs of meningococcal disease in Italy; only partial data on hospitalization costs were available, therefore we analyzed the data from studies conducted in other high-income countries. A total of 23 articles were included in the present overview.

Direct costs refer to those borne by the National Health Service (NHS), and are divided into acute-phase, post-acute-phase, long-term-phase costs.

The direct costs of the acute phase include the costs of hospitalization, rehabilitation and the public health response. Table 2 reports the direct costs associated with acute-phase disease related to the hospital discharge forms (HDF).

In Italy, the mean direct costs of hospitalization range from EUR 4529 to EUR 6708 per case according to the patient’s clinical conditions and age, on the basis of the Hospital Discharge Form codes (ICD-9) assigned to meningococcal meningitis, septicemia and both of these clinical conditions, and the related Diagnosis Related Groups (DRG) codes [36].

Detailed data on direct costs related to the public health response (number of contacts requiring antibiotic chemoprophylaxis, any vaccination campaign undertaken and the mean time spent by the staff of the Local Health Agency to avoid secondary cases) are not yet available in Italy. Nevertheless, a preliminary assessment has estimated that the costs of the public health response amount to about EUR 3284 per case [25,37,38].

The post-acute phase direct costs include the costs of managing sequelae, rehabilitation and psychiatric and psychological support for the patient during the six months that follow the acute phase of the disease. These costs vary according to the type of sequelae: for example, in Italy, the direct cost of each outpatient examination is EUR 20.66, and the cost of each session of psychological support is EUR 19.37 [25,36].

The overall costs incurred during the long-term phase are very variable and depend on the type of sequela. For example, a recent Health Technology Assessment (HTA) report [25], aimed at evaluating anti-meningococcal B vaccination with the vaccine Trumenba^®^ in adolescents in Italy, evaluated the annual long-term direct cost broken down by sequelae; these were: EUR 2463.67 for amputation with substantial disability; EUR 2068.55 (first year) and EUR 1086.25 (subsequent years) for severe skin damage; EUR 10394.80 (first year) and EUR 4345.38 (subsequent years) for renal dysfunction; EUR 96682.72 for severe neurological damage; EUR 757.58 for visual disorders, etc.

Indirect costs also have a high social impact, as they include, for example, those related to the loss of productivity of patients (EUR 27,700 for a single case) and their family members (EUR 13,850 for a single case) [25], and the costs of managing patients with severe sequelae, specifically, the costs of special education (EUR 14,842.75 for a single case) [25], disability pensions, invalidity benefits and accompaniment allowances [25,37,38,39,40,41].

### 3.3. Immunogenicity and Safety of Anti-Meningococcus B Vaccines (MenB-4C and MenB-FHbp)

The development of a broadly protective Men B vaccine has required great research efforts, and only in 2013 did a quadrivalent Men B vaccine for subjects aged ≥ 2 months of life become available. This vaccine (MenB-4C) contains three recombinant antigen proteins – Factor H binding protein, *Neisseria* adhesion A and Neisserial Heparin Binding Antigen - (FHbp, NadA and NHBA) and outer membrane vesicles (OMV), as amount of total protein containing PorA P1.4 [2,23].

Immunogenicity against the four antigens is high in adolescents (≥11 years of age) after a 2-dose schedule: 100% of seropositivity rates against FHbp, NHba and PorA P1.4, and 99–100% against NadA. The high level of seropositivity for all the antigens persists even 18–23 months after the administration of the last dose. After a period of 7.5 years following the primary immunization, the antibody protection tends to decline, but the administration of a booster dose elicits a seroconversion against all the antigens in 93–100% of subjects, suggesting the presence of an immunological memory [23]. The MenB-4C vaccine can be administered concomitantly with the following vaccines (in either the monovalent or combined form): diphtheria, tetanus, acellular pertussis, *Haemophilus influenzae* type b, inactivated poliomyelitis, hepatitis B, heptavalent pneumococcal conjugate, measles, mumps, rubella, varicella, and meningococcal groups A, C, W, Y conjugate (MenACWY) [23]. This vaccine is safe and well tolerated: among adolescents and adults, the most common local and systemic adverse reactions are generally pain at the injection site, malaise and headache [23].

In 2017, the Italian Medicines Agency licensed a new vaccine (MenB-FHbp) for the active immunization of subjects aged ≥10 years. This recombinant vaccine contains two variants of the complement factor H-binding protein (FHbp) and is adjuvated with a lipidic component. More than 96% of the meningococcal B strains isolated in Europe express these two variants, A and B, of FHbp on their surface [24].

Immunization with the MenB-FHbp vaccine stimulates the production of bactericidal antibodies that recognize FHbp expressed by meningococcus [24]. The results obtained in several phase II and phase III clinical studies, involving adolescents (10–18 years) and young adults (19–25 years), have shown that the MenB-FHbp vaccine elicits a broad immune response against antigenically different strains of meningococcus B, after either a 2-dose or 3-dose schedule [24,42,43,44] Following a primary immunization course, the antibody titers persists up to 4 years after the vaccination [24]. Furthermore, the administration of a single dose of vaccine 4 years after the primary immunization has been seen to elicit a strong immune response, suggesting the importance of administering a dose booster in order to guarantee an adequate level of protection over time [24,45,46]. The MenB-FHbp vaccine can be co-administered with other vaccines recommended in adolescence, such as the tetravalent anti-human papillomavirus vaccine (HPV4), the anti-diphtheria-tetanus-acellular pertussis-inactivated poliovirus vaccine (DTaP-IPV) and the MenACWY vaccine [47,48].

The bivalent anti-MenB vaccine is safe and well tolerated. Adverse reactions are generally mild or moderate, the most common being pain, redness and swelling at the injection site, headache, fatigue, chills, diarrhea, muscle pain, joint pain and nausea [24].

The characteristics of the two available vaccines are summarized in Table 3

### 3.4. The Cost-Effective Profile of an Immunization Program against Meningococcal B Targeted to the Adolescent Population in Italy

A recent HTA report assessed the impact of the implementation of an immunization program with the MenB-FHbp vaccine for adolescents (aged 11 years old) in terms of cost–utility in the Italian context by applying a Markov cohort simulation model [25]. The model compared the “vaccination strategy” with the “non-vaccination strategy”, the latter being the current standard of care (as the NVPP 2017–2019 [17] does not include the anti-meningococcal B vaccination in adolescence). The 2-dose vaccination schedule was considered in the model. The results showed that the “vaccination strategy” was cost-effective. Indeed, the ICER was EUR 7911.98/QALY from the NHS perspective, and EUR 7757.73/QALY from the perspective of society [25], considering a value of EUR 30,000 as the cost-effectiveness threshold [49].

No study has analyzed the economic profile of a vaccination strategy for adolescents with the MenB-4C vaccine in Italy.

## 4. Discussion

The above evidence regarding the epidemiology and burden of meningococcal B disease in adolescents, its related costs, the availability of immunogenic and safe vaccines and the cost-effectiveness of vaccination against *N. meningitis* B indicate that this vaccination should, in the near future, be included among those recommended for adolescents in Italy.

Although cases of invasive meningococcal disease are relatively uncommon, the disease has a severe course, a high lethality rate and a high probability of causing invalidating sequelae. Moreover, considering all the direct and indirect costs of the disease and its sequelae, it is possible to conclude that each case generates significant costs for the NHS and for society. For example, the mean overall acute-phase cost of a single case amounts to about EUR 13,952; the cost accruing to the post-acute and the long-term phases may vary widely according to the type of sequela, reaching an annual cost of about EUR 100,000 in the case of severe neurological damage [25]. It is important to note that, to date, no exhaustive studies have evaluated in detail the sequelae of meningococcal disease and their global costs in Italy. In this context, it therefore seems necessary to design Italian studies to evaluate the consequences of meningococcal disease broken down by age-groups in the near future.

Adolescents are a major source of transmission of the micro-organism, as they often frequent crowded places of social aggregation (e.g., schools, universities and pubs), as revealed by some recent outbreaks [50]. In 2018, seven cases of meningitis caused by *N. meningitidis* B were reported in Sardinia (Italy) in young people who had attended a nightclub [51], none of whom had been vaccinated.

It is very important to note that many adolescents are carriers, and that healthy carriers are the main source of the disease; indeed, secondary cases due to contagion from a sick individual are very few [25]. The prevalence of healthy carriers varies according to age: from 4–5% in children to >20% in young people, with a peak around the age of 20 years, and then declining in the older population [15,52]. Considering the Italian context, a recent study conducted in Milan revealed that 5.3% of the population aged 14–21 years were healthy carriers [16]. Another study, conducted in Genoa (study population: subjects aged 14–22 years) reported a higher value, i.e., 18.5% subjects were found to be carriers of *N. meningitidis* with the highest rate of carriage observed at 17 and 18 years of age (25.0% and 25.5%, respectively) [14]. In both Italian studies, strains belonging to serogroup B were the most common [14,16].

To date, despite the heavy clinical, social and economic burden of disease in adolescents and their role as carrier, in Europe there are no general recommendations regarding the administration of the anti-MenB vaccine for this age-group [53]; in the United States, by contrast, the Advisory Committee on Immunization Practices (ACIP) strongly recommends immunizing adolescents and young adults, aged between 16 and 23 years, against meningococcus B [54].

Some research groups are currently evaluating the benefits that could result from a free MenB vaccination program for adolescents. In Italy, a recent HTA report assessed the impact of immunizing adolescents (aged 11 years old) with the MenB-FHbp vaccine; the authors demonstrated that, if the overall impact of the disease is adequately considered, this strategy would be strongly recommendable, even in the case of a low incidence disease. They concluded that vaccination would determine not only a reduction in the number of cases of disease, but also savings in the direct and indirect costs related to the permanent and invalidating sequelae that afflict a considerable number of survivors. Indeed, the study revealed that the long-term costs have the greatest impact on the level of cost-effectiveness [25].

Another study, conducted in Canada, reported that, if the vaccine reduces risk of carriage acquisition, vaccination of older adolescents, even at lower uptake, could have a significant public health impact, and that evaluations of new programs should consider the overall benefits of the vaccination [55].

In the UK, a dynamic transmission model was developed with the objective of assessing the impact of MenB-4C and/or MenACWY vaccination strategies on invasive meningococcal disease. In comparation with no vaccination, the smallest reduction in meningococcal cases was observed for MenACWY strategies (toddler and/or adolescent); while MenB-4C (infant or infant/adolescent), alone or with MenACWY, always resulted in the most rapid and steep decline in cases. Combined strategies involving adolescent MenB-4C resulted in the largest decrease in cases. Supplementing MenB-4C infant and MenACWY adolescent programs with a MenB-4C adolescent strategy would achieve the largest overall reduction in IMD cases [56].

When a new vaccination program is to be implemented, effective strategies to achieve adequate vaccination coverage should be evaluated, including suitable information and communication campaigns.

A factor strictly associated with adherence to an immunization program is the diffusion of adequate information about available vaccines, their efficacy and their safety. In the last decade, the access to health information has become easier for people thanks to the web and social networks; but the contents are not always correct, of high quality and complete. Certainly, the internet and social networks are useful channels to disseminate health information, especially among adolescents and young adults (great users of the media), but it is essential to make the population aware about the risks and the unreliability of information in some websites. Considering this aspect, new integrated communication strategies should be evaluated to highlight the message of the importance of the vaccination to adolescents and their parents. Adolescents particularly need to be included in the process of the informed choice, in order to give them the possibility to make choices about their health. The implementation of targeted strategies would make it possible to achieve high levels of compliance with meningococcal B vaccination.

This new immunization program could be favored by the fact that it involves a severe disease, and therefore the perceived risk of the disease is high, especially among parents who are very concerned about their children’s health and among adolescents who are aware of their high risk of contracting the meningococcal disease. For this reason, the acceptability of an anti-meningococcal vaccination is higher compared to other vaccinations [57,58,59,60,61,62]. As a matter of fact, it is known that the public’s perception of the risk of contracting a vaccine-preventable disease and of suffering due to its consequences is a factor that influences the adherence to a vaccination program [63,64,65,66].

Moreover, in order to increase public awareness, it is essential to strengthen a constructive collaboration among public health workers, family pediatricians and general practitioners.

To overcome the lack of effective communication about the availability of a vaccination, not only ordinary letters should be sent, but also text messages or emails should be considered as new reminder methods [67,68,69]. Some studies have already highlighted the effectiveness of this approach, making it possible to reach large numbers of subjects [70].

Concerning the organizational aspects, the anti-MenB vaccine could be to co-administered with the other vaccines administered during adolescence, such as anti-HPV, anti-MenACYW135, Tdap-IPV and Tdap vaccines [24]. Since the nonavalent anti-HPV vaccine has been recently licensed, no data are yet available for its co-administration with meningococcal B vaccine. However, knowing that the development process of the nonavalent anti-HPV is similar to the quadrivalent one, it is possible to suppose a co-administration [71,72]. This could increase the compliance for all vaccines. For example, Sicily, one of the three pilot Regions with meningococcal B immunization, has adopted this strategy [19].

Focusing on the organization of the vaccination centers, the allocation of adequate financial resources would make it possible to extend the opening hours and to organize the vaccination session in “strategic” settings, such as clubs attended by young people, or schools [73,74]. The Italian health authorities should consider the possibility of administering the anti-MenB vaccination in the school setting in order to optimize the entire organizational process. This is not a new for Italy, in fact in the past, some vaccination campaigns have been successfully carried out in schools. Several studies have already provided evidence of the effectiveness of this approach in increasing both the adherence to vaccination and the awareness, resulting in a positive impact on the general population [75,76,77,78,79]. Furthermore, this would improve accessibility to the vaccination services and make it easier to involve adolescents who are generally “hard to reach” subjects [75].

Finally, considering the ethical aspect, vaccinating adolescents against Men B would safeguard their lives and health, avoiding the occurrence of severe lifelong sequelae, as shown by the implementation of a “triangular model”, based on a person-centered approach [25,80].

## 5. Conclusions

In Italy, the meningococcal B disease has a low incidence, but the burden of the disease is very heavy due to the high lethality rate and high frequency of transitory and/or permanent sequelae among survivors, especially in children and adolescents. Moreover, the latter are the main healthy carriers and can be the major source of infection due to their social behaviors. Due to the severity of the disease, the direct and indirect costs of each single case are high both for the NHS and for society. Indeed, the premature death and the impaired quality of life of survivors with sequelae and their relatives/caregivers have a strong impact.

Actually, we have immunogenic and safe vaccines that can be administered through a cost-effective strategy; therefore, it is now time to decide at a national level on the introduction of the anti-meningococcal B vaccination during adolescence. If this new immunization program will be implemented, ensuring equality of treatment at a national level, the fight against the invasive meningococcal disease will be further strengthened.

## Figures and Tables

**Table 1 microorganisms-08-01681-t001:** The most frequent types of sequelae (physical, neurological and/or psychological), reported in the literature.

Type of Sequelae	Probability	Ref.
Physical sequelae		
Skin scars	6.4–48%	[27]
Amputations	0.8–14%	[27]
Renal dysfunction	2–8.7%	[27]
Arthritis/vasculitis	4.7%	[27]
Limb deformity	6%	[7]
Neurological sequelae		
Bilateral/unilateral deafness	up to 19%	[1,25,26,27,28,29,30,31]
Cognitive impairment	up to 24%	[1,25,26,27,28,29,30,31]
Convulsions/epilepsy	up to 40%	[1,25,26,27,28,29,30,31]
Visual disorders	up to 23%	[1,25,26,27,28,29,30,31]
Problems of communication	up to 25%	[1,25,26,27,28,29,30,31]
Psychiatric and psychological problems		
Depression	5.7%	[31]
Anxiety	7.1%	[31]

**Table 2 microorganisms-08-01681-t002:** The direct costs associated with acute-phase disease.

Parameter	Cost (EUR)	Ref.
Hospitalization (subjects <18 years)	HDF-9 code 036: € 4952.64 ^a^ HDF -9 code 0360: € 4952.64 ^b^ HDF -9 code 0362: € 4529.00 ^c^	[36]
Hospitalization (subjects >18 years)	HDF -9 code 036: € 6708.04 ^a^ HDF -9 code 0360: € 6542.06 ^b^ HDF -9 code 0362: € 6177.88 ^c^	[36]
Public health response	€ 3284	[36]
Hospitalization in long-term care (maximum 20 days)	€ 4040.00 (€ 202,00/day)	[36]

^a^ = mean of the principal DRGs associated with HDF-9 code 036. ^b^ = mean of the principal DRGs associated with HDF-9 code 0360. ^c^ = mean of the principal DRGs associated with HDF-9 code 0362 (DRG: Diagnosis Related Groups).

**Table 3 microorganisms-08-01681-t003:** Summary of the characteristic of the anti-meningococcus B vaccines licensed in Italy, referring to the adolescent population. NOTE: *:% seropositivity (% of subjects who achieved an hSBA ≥ 1:4) one month after the last dose. hSBA: serum bactericidal activity with human complement. §:% of subjects who achieved a ≥ 4-Fold rise in hSBA titre one month after the last dose.

Anti-Meningococcus B Vaccines Licensed in Italy: Recommendation for Adolescents.		
	MenB4-C [23]	MenB-FHbp [24]
**Composition**	Three recombinant antigen proteins (FHbp, NadA and NHBA) and outer membrane vesicles (OMV) as amount of total protein containing PorA P1.a	Two recombinant lapidated antigen variants of FHbp protein (subfamily A and B).
**Indications**	Active immunization in subjects aged ≥ 2 months.	Active immunization in subjects aged ≥10 years.
**Posology**	Primary immunization: 2-dose schedule (0, 1 or 0, 2 or 0, 6 months). Booster is recommended for high-risk group.	Primary immunization: 2-dose schedule (0, 6 months) or 3-dose schedule (0, 1, 6 or 0, 2, 6 months). Booster is recommended for high-risk group.
**Immunogenicity**	FHbp, NadA, PorA P1.4: 100% NHBA: 99–100% (*)	A22: 78.1–84%; A56: 93.4–94.2% B24: 74.6–75.4%; B44: 81.7–82.2% (§).
**Safety**	Most common: pain at the injection site, malaise, headache.	Most common: pain, redness and swelling at the injection site, headache, fatigue, chills, diarrhea, muscle pain, joint pain.
**Co-administration**	Monovalent or combined vaccines: diphtheria, tetanus, acellular pertussis, Hib, IPV, HBV, heptavalent pneumococcal conjugate, measles, mumps, rubella, varicella, and meningococcal groups A, C, W, Y conjugate.	Tetanus Toxoid, Reduced Diphtheria Toxoid, Acellular Pertussis, and Inactivated Poliovirus Vaccine (TdaP-IPV), Quadrivalent Human Papillomavirus vaccine, meningococcal groups A, C, W, Y conjugate vaccine and Tetanus Toxoid, Reduced Diphtheria Toxoid, and Acellular Pertussis Vaccine Adsorbed (Tdap).

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
