# Peer review of "Why the Anti-Meningococcal B Vaccination during Adolescence Should Be Implemented in Italy: An Overview of Available Evidence"

_microorganisms, 2020, doi:10.3390/microorganisms8111681_

Round 1
Reviewer 1 Report
This paper review describes the status of meningococcal disease burden in Italy and suggests that anti-meningococcal B vaccination is necessary during the adolescence period. Although anti-meningococcal B vaccination might be important in Italy in current status, some parts of this review article did not describe and explain well to convince us to accept these suggestions. There are some points which need further explanation and clarification.
Major comments :
- The basic introduction of meningococcal disease and how important of this disease in Italy even worldwide should be properly addressed.
- In the abstract, the authors described that the incidence rate of meningococcal disease in Italy is low but the burden of the disease is heavy. These descriptions were conflicted. Please try to state more clear and clarify it.
- The part of "materials and methods" section should be revised. It is suggested to describe how to collect the data, select data, rule out some improper references ……Normally, a good review should be followed the PRISMA guidelines to summarize and describe each topic logistically. Please revise the materials and methods according to PRISMA.
- Some descriptions in the materials and methods should be moved to the Results and cited properly refs.
- 1 to 3 summery tables are suggested to give, which briefly summarize each point that describe in the Results section and also summarize the related refs corresponding to each statement. Such as current vaccine types , cost-effective profile of each vaccine in the immunization program etc.
- The references cited in this paper should double check and confirm. In addition, as the description “To date, in Europe there are no general recommendations regarding the administration of the anti-MenB vaccine during adolescence”. This may due to low incidence and very a few cases reported in Europe. The authors should compare and discuss their rationale regarding the importance of vaccination and why the vaccination program or policy are different between Italy and other European countries and even Europe and other countries.
Author Response
"Please see the attachment.

Reviewer 2 Report
This paper can contribute to a greater implementation of vaccination against meningococcal B.
I think that the paper should be improved in some aspects as:
- To expand a little the introduction section. The impact of this disease should be included.
- Results. In order to follow better the results I think that some of them should be put in table form. In this way the reader will be more aware of the different aspects of the work and their meaning.
Round 2
Reviewer 1 Report
This review article has been greatly improved and most questions were addressed. However, there were some points which need further confirm and clarification.
- Page 1, line 69-70, “The only weapon available in the fight against invasive disease due to Neisseria meningitidis is vaccination”. Please clarify this sentence. Why the authors feel that vaccine is only weapon? how about anti-microbial therapy? or other new development agent ?
- Line 144-145, “We researched cost-effectiveness studies of anti-MenB vaccination in adolescents in Italy in order to evaluate the economic profile of the new immunization programme” Where is the data source ? or reference ?
- Line 173 “To date, no study has analyzed the impact of sequelae due to meningococcal disease in Italy”. Please confirm this description, is it the truth that this study is the only one to analyze this problem. Because this article is a Review instead of a research article. If no other report or reference, how come the authors could review this issue. Similar question was shown in Line 215 “ To date, no exhaustive studies have evaluated the global costs of meningococcal disease in Italy. Please clarify these descriptions.
- Line 143, Cost-effectiveness analyses of anti-MenB vaccination in adolescents. Is there any reference selected for analysis or comparison for this issue ?
- The English editing is still necessary to make this article to be read smoothly.
